# Investigation on Modeling and Formation Mechanism of Dynamic Rotational Error for Spindle-Rolling Bearing System

**Gaofeng Hu** [1,2,*], **Ye Chen** [3], **Liangyu Cui** [1] 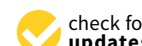, **Gang Jin** [1], **Tingjian Wang** [1], **Houjun Qi** [1,4] **and Yanling Tian** [2]

1   School of Mechanical Engineering, Tianjin University of Technology and Education, Tianjin 300222, China; cuily@tju.edu.cn (L.C.); jgang1982@163.com (G.J.); tjwang@hit.edu.cn (T.W.); qihoujuntj@163.com (H.Q.)
2   School of Mechanical Engineering, Tianjin University, Tianjin 300350, China; meytian@tju.edu.cn
3   Beijing Institute of Aerospace Control Devices, Beijing 100854, China; chenye@tju.edu.cn
4   Tianjin Key Laboratory of High Speed and Precision Machining, Tianjin 300222, China
*   Correspondence: gaofenghu@tju.edu.cn

**Abstract:** In the field of precision machining, the spindle-rolling bearing (SRB) system is widely used on the machine tool as one of the most fundamental and important components. The rotational error motions of the SRB system have significant effects on the machining accuracy (contour accuracy and surface roughness). Over the past decades, much work has been focused on the measurement of spindle balancing and rotational error motions, the vibrations response induced by the nonlinear stiffness and surface waviness of the bearing. However, the formative mechanism of the rotational error motions for the SRB system is not well understood. In this paper, the dynamic model of the SRB system considering the bearing nonlinearity is established. Seeking to reveal the effects of surface waviness of the bearing raceway, unbalance mass and disturbance force on the dynamic rotational error, the modeling method and formative mechanism of the dynamic rotational error for the SRB system is explored both theoretically and experimentally. Then, numerical simulation is performed to analyze the influence of the bearing raceway waviness, unbalance mass and disturbance force on the dynamic rotational error. An experimental setup is established based on a typical SRB system and a series of experiments are carried out. The experimental results are in good agreement with the theoretical and simulation results, which can demonstrate the feasibility and validity of the modeling method. Furthermore, this method can be effectively applied to the design and development phases of an SRB system to improve dynamic rotational accuracy.

**Keywords:** spindle-rolling bearing system; dynamic rotational error; nonlinear bearing model; surface waviness; dynamic unbalance; disturbance force

## 1. Introduction

The spindle-rolling bearing (SRB) system is a key component of the machine tool or other rotating machines, which plays a key role in working. The rotational accuracy is a critical technical indicator for the SRB system when evaluating the dynamic performance [1,2]. With the increasing demands on machining accuracy of the machine tool, much effort has been made to research the measuring methods on the rotational accuracy of the SRB system. Metrologists began to measure rotational accuracy as early as the 1900s. Scheslinger [3] used a mechanical indicator to measure the total indicator reading (TIR), which is recognized to be a static measuring method. Tlusty [4] and Bryan [5] pioneered modern measuring methods by using more accurate data acquisition (DAQ) system and evaluating methods. Their methods lay the foundation for ANSI(American National Standards

Institute)/ASME(American Society of Mechanical Engineers) and ISO (International Standardization Organization)testing standards [6,7], which are commonly used for testing machine tool spindle rotational accuracy nowadays. In order to obtain more accurate test results, researchers developed nanometer-resolution capacitive displacement sensors and a corresponding spindle error analyzer. A long series of ingenious error separation methods and corresponding measuring setups have been proposed to improve the measuring accuracy [8–14]. Chen [15] presents an identification method for the spindle rotation error in the workpiece surface by the wavelet transform, Weierstrass function and power spectral density. A commercial spindle error analyzer is developed by the Lion Precision company, as well as high-precision displacement sensors [16].

It is well known that some key factors including the Hertz contacts, bearing preload, surface waviness, unbalance mass and so on, would significantly affect the dynamic rotational error of the SRB system. These factors can significantly decrease the spindle rotational accuracy. Huang et al. [17], established a general rotating error model according to the structure characteristic of SRB system. In the model, the effects of source errors and the action rule were analyzed. However, different kinds of source errors are just superimposed in accordance with their geometric relation, which can be only used for static analyses. Huang and Lee et al. investigated the effects of spindle unbalance on the machining accuracy of the aerostatic bearing spindle (ABS). A dynamic model of the ABS was proposed to analyze the error motion and corresponding dynamic characteristics. Besides, in order to evaluate the spindle accuracy, simple frequency analyses were adopted [18]. Kim and Igor Zverv et al. [19], studied the spindle rotational error motion caused by non-ideal ball bearing by establishing a dynamic model of SRB system. Multiple error patterns were analyzed in the model. To evaluate the rotational error motion, root-mean-square summation of harmonics was calculated by using the displacement data of spindle vibration. Harsha et al. [20,21] presented an analytical model to investigate the nonlinear dynamic response of the SRB system, with consideration of the surface waviness, cage runout, number of balls, surface waviness and radial internal clearance. G. Jang and Jeong [22] also presented a nonlinear model to analyze the ball bearing vibration due to the waviness in a rigid rotor supported by ball bearings. Bai and Xu [23] present a general dynamic model for studying the dynamic properties of the rotor system supported by ball bearings under the effects of both internal clearance and bearing running surface waviness, and conclude that the clearance, axial preload and radial force play a significant role in affecting the system stability. Q. K. Han and F. L. Chu [24,25] established a nonlinear dynamic model based on the Hertz contact theory, with the purpose of investigating the effects of axial preload, spindle eccentricity, inner/outer waviness amplitudes on the dynamic response in both time domain and frequency domain. Xiaohu Li [26], discussed the mathematical description and evaluation method on spindle radial error motion, the axial thermal displacement, radial error motion and modal characteristic of spindle were verified by the designed experiment. Some researchers [27–30] investigated the vibration response induced by the nonlinear stiffness, surface waviness and distribute defects of the bearing. It is helpful to the study of running accuracy of the spindle; however, it is insufficient to investigate the modeling method and formative mechanism of the dynamic rotational error motion.

From the literature reviews, it is worth noting that most research studies have been focused on how to measure the rotational accuracy of the SRB system and how to improve the measuring accuracy and the vibrations response induced by the nonlinear stiffness and surface waviness of the bearing. However, few papers are published associating these error motion sources with the spindle rotational accuracy, especially for dynamic rotational accuracy. The modeling and formative mechanism of the dynamic rotational error motion are not well understood. Motivated by this, this paper mainly focuses on the modeling method and the formative mechanism of spindle rotational error motion. The structure of this paper is arranged as follows: Section 2 introduces a dynamic model of the spindle-rolling bearing (SRB) system considering the bearing nonlinearity to analyze the effecting mechanism of the error factors. Based on the model of Section 2, the influence of the error factors is analyzed quantitatively by utilizing the numerical simulation in Section 3. In Section 4, an experimental

setup is designed to verify the proposed model. A series of measuring experiments for the dynamic rotational error corresponding to different influenced factors are performed. The conclusions and prospects of this study are given in Section 5.

## 2. Modeling of Dynamic Rotational Error for Spindle-Rolling Bearing System

Angular contact ball bearings are widely used in the SRB system. The nonlinear and time-varying characteristics of the bearing, under the influence of geometric error, bearing structure and dynamic load, is the root cause of the nonlinearity and has great important effects on the dynamic rotational error of the SRB system. In this section, the dynamic model of the SRB system considering the nonlinear and time-varying behaviors of the bearing are established, based on which the dynamic rotational error is evaluated.

### 2.1. Nonlinear Bearing Model

The dynamic supporting stiffness and time-varying bearing force are important parameters in modeling the nonlinear behaviors of rolling bearings. During operation, the dynamic supporting stiffness of the SRB system is nonlinear according to the Hertz contact theory (the Hertz contact theory illustrates the distribution rule of local stress and strain generated by the contact of two objects under compression), while the bearing force is time-variable due to the machining error of the bearing rings and rolling elements.

### 2.1.1. Modeling of Dynamic Supporting Stiffness

The geometry of an angular contact ball bearing and the coordinate systems is shown in Figure 1. The global coordinate system (O-XYZ) is fixed at the rotating axis center of the outer ring. The inner ring rotates at a constant speed $\omega_0$ with Z axis while the outer ring is fixed. $\alpha_o$ is the initial contact angle of the bearing. The outer ring is fixed and assume that $\delta_x$, $\delta_y$, $\delta_z$, $\theta_x$ and $\theta_y$ are the three translational and two angular displacements of the inner ring. The displacement vector of the inner ring is expressed as

$$\delta_i = \left[\delta_x, \delta_y, \delta_z, \theta_x, \theta_y\right]^T,\tag{1}$$

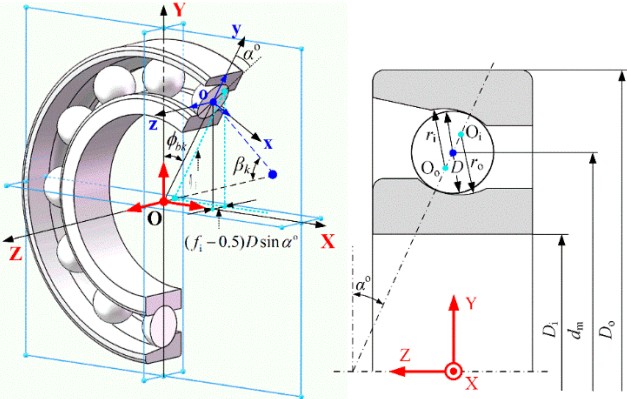

**Figure 1.** The structure of the angular contact ball bearing.

There are $N_b$ rolling elements in the bearing and the position angle of the kth rolling element, $\phi_{bk}$, could be obtained from the following formula:

$$f_{bk} = \frac{2\pi(k-1)}{N_b},\tag{2}$$

The local coordinate system (o-xyz), whose center is at the mass center of the rolling element, rotates at an angular speed $\omega_{bk}$ with the Z axis. $\omega_{bk}$ is the orbital angular speed of the rolling element and can be expressed as

$$\omega_{bk} = \frac{\omega_0}{2}\left(1 - \frac{D}{d_m}\cos\alpha^o\right), \tag{3}$$

where $d_m$ is the pitch diameter of the bearing, D is the diameter of the rolling element.

The rolling elements also rotate at an angular speed $\omega_{sk}$ around the axis of itself, which is called the spinning axis. The self-rotation angular speed $\omega_{sk}$ can be easily obtained by [31].

$$\omega_{sk} = \frac{d_m\omega_0}{2D}\left(\left(\frac{D}{d_m}\cos\alpha^o\right)^2 - 1\right), \tag{4}$$

The angle between the spinning axis of the rolling elements and Z axis is $\beta_k$, which is called as rolling element attitude angle. $\beta_k$ can be expressed as [31].

$$\beta_k = \tan^{-1}\left(\frac{\sin\alpha_{ok}}{\cos\alpha_{ok} + \frac{D}{d_m}}\right), \tag{5}$$

When the rotational speed $\omega_0$ and the axial preload are zero, both the inner contact angle and the outer contact angle are the same and equal to the initial contact angle $\alpha^o$. The inner raceway groove curvature center, outer raceway groove curvature center and the rolling element center are collinear (red line), as shown in Figure 2. The distance between the inner raceway groove curvature center and the rolling element center, as well as the distance between the outer raceway groove curvature center and the rolling element center before deformation are

$$\begin{aligned}L_{ik} = r_i - 0.5D = (f_i - 0.5)D\\L_{ok} = r_o - 0.5D = (f_o - 0.5)D\end{aligned}\ ' \tag{6}$$

where $r_i$ and $r_o$ are the inner and outer raceway groove curvature radius, respectively, $f_i$ and $f_o$ are the inner and outer raceway groove curvature radius coefficient, respectively.

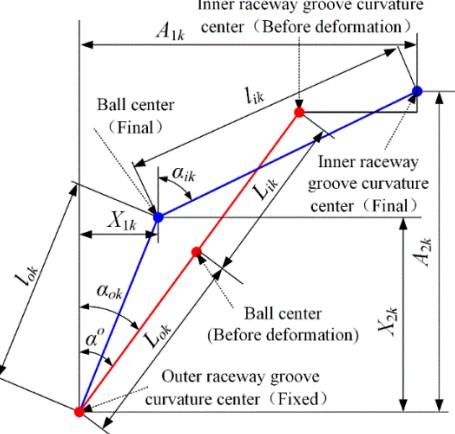

**Figure 2.** The relative positions of the kth rolling element center with the inner and outer race groove curvature centers.

With the rotational speed $\omega_0$ and axial preload increase, the inner raceway groove curvature center, outer raceway groove curvature center and the rolling element center are not collinear (blue line). The inner contact angle and the outer contact angle are not equal to each other. The inner contact angle increase while the outer contact angle decreases due to the centrifugal force. Besides, the distance between the inner/outer raceway groove curvature center and rolling element center changes in the

Y–Z plane. The distance between the inner raceway groove curvature center and the rolling element center, as well as the distance between the outer raceway groove curvature center and the rolling element center, after deformation are

$$
\begin{aligned}
l_{ik} &= (f_i - 0.5)D + \delta_{ik} \\
l_{ok} &= (f_o - 0.5)D + \delta_{ok}
\end{aligned}
, \tag{7}
$$

where $\delta_{ik}/\delta_{ok}$ is the contact deformation between the inner/outer bearing ring and the kth rolling element.

In accordance with the displacement coordinate relations, the axial distance between the inner and outer raceway groove curvature centers is

$$
\begin{aligned}
A_{1k} &= (L_{ok} + L_{ik})D\sin\alpha^0 + \delta_z + \eta_i\theta_x\sin f_{bk} - \eta_i\theta_y\cos f_{bk} \\
A_{2k} &= (L_{ok} + L_{ik})D\cos\alpha^0 + \delta_x\cos f_{bk} + \delta_y\sin f_{bk}
\end{aligned}
, \tag{8}
$$

where $\eta_i$ is the distance between the rotating axis Z and inner raceway groove curvature center and can be calculated by

$$
\eta_i = \frac{d_m}{2} + (f_i - 0.5)D_b\cos\alpha_0, \tag{9}
$$

It convenient to introduce new variables $X_{1k}$ and $X_{2k}$, as shown in Figure 2. It can be seen from Figure 2 that at any rolling element location

$$
\begin{aligned}
\sin\alpha_{ik} &= \frac{X_{1k}}{l_{ik}} \\
\cos\alpha_{ik} &= \frac{X_{2k}}{l_{ik}} \\
\sin\alpha_{ik} &= \frac{A_{1k} - X_{1k}}{l_{ik}} \\
\cos\alpha_{oik} &= \frac{A_{2k} - X_{2k}}{l_{ik}}
\end{aligned}
\tag{10}
$$

Using the Pythagorean Theorem, it can be seen from Figure 2 that [31].

$$
\begin{aligned}
(A_{1k} - X_{1k})^2 + (A_{2k} - X_{2k})^2 - [(f_i - 0.5)D + \delta_{ik}]^2 &= 0 \\
X_{1k}^2 + X_{2k}^2 - [(f_o - 0.5)D + \delta_{ok}]^2 &= 0
\end{aligned}
\tag{11}
$$

As shown in Figure 3, the load equilibrium equations for the kth rolling element can be expressed as [31].

$$
\begin{aligned}
Q_{ik}\cos\alpha_{ik} - Q_{ok}\cos\alpha_{ok} + \frac{\lambda_{ik}M_{gk}}{D}\sin\alpha_{ik} - \frac{\lambda_{ok}M_{gk}}{D}\sin\alpha_{ok} + F_{ck} &= 0 \\
Q_{ik}\sin\alpha_{ik} - Q_{ok}\sin\alpha_{ok} - \frac{\lambda_{ik}M_{gk}}{D}\cos\alpha_{ik} + \frac{\lambda_{ok}M_{gk}}{D}\cos\alpha_{ok} &= 0
\end{aligned}
\tag{12}
$$

where $Q_{ik}/Q_{ok}$ is the contact force between the inner/outer ring and the rolling element. $Q_{ik}$ and $Q_{ok}$ can be expressed as

$$
\begin{aligned}
Q_{ik} &= K_{ik}\delta_{ik}^{1.5} \\
Q_{ok} &= K_{ok}\delta_{ok}^{1.5}
\end{aligned}
\tag{13}
$$

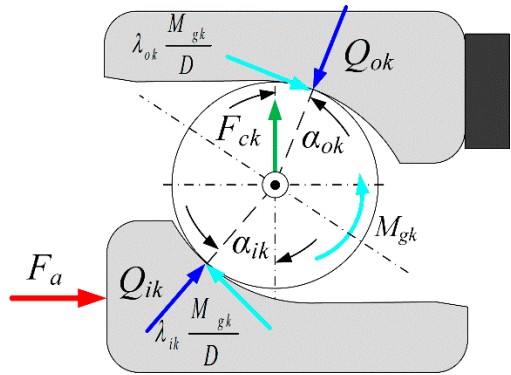

**Figure 3.** Forces exerted on the jth rolling element.

$\lambda_{ik}/\lambda_{ok}$ represent the friction coefficients between the inner/outer raceway and rolling element. As for the outer raceway control, $\lambda_{ik} = 0$ and $\lambda_{ok} = 2$ The gyroscopic moment $M_{gk}$ and centrifugal force $F_{ck}$ acting on the kth rolling element can be described as follows:

$$M_{gk} = I_b\omega_{bk}\omega_{sk}\sin\beta_k$$
$$F_{ck} = 0.5m_b d_m \omega_{bk}^2 \tag{14}$$

where $m_b$, $I_b$ is the mass and mass moment inertia of the kth rolling element, respectively.

Equations (11) and (12) are highly coupled and nonlinear and can be resolved by Newton–Raphson method. Then the contact forces $Q_{i:}$, $Q_{o:}$, contact angles $\alpha_{i:}$, $\alpha_{o:}$, and gyroscopic moments $M_{i:}$ $M_{o:}$, between each rolling element and inner/outer rings can be obtained. The resultant force vector of the inner ring due to all of the contacting rolling elements is

$$F_i = \left[F_x, F_y, F_z, M_x, M_y\right]^T \tag{15}$$

where

$$F_x = \sum_{k=1}^{N_b}\left(Q_{ik}\cos\alpha_{ik} + \frac{\lambda_{ik}M_{gk}}{D}\sin\alpha_{ik}\right)\sin f_{bk}$$

$$F_y = \sum_{k=1}^{N_b}\left(Q_{ik}\cos\alpha_{ik} + \frac{\lambda_{ik}M_{gk}}{D}\sin\alpha_{ik}\right)\cos f_{bk}$$

$$F_z = \sum_{k=1}^{N_b}\left(Q_{ik}\sin\alpha_{ik} - \frac{\lambda_{ik}M_{gk}}{D}\cos\alpha_{ik}\right) \tag{16}$$

$$M_x = \sum_{k=1}^{N_b}\left(Q_{ik}\sin\alpha_{ik} - \frac{\lambda_{ik}M_{gk}}{D}\cos\alpha_{ik}\right)\eta_i\cos f_{bk}$$

$$M_y = -\sum_{k=1}^{N_b}\left(Q_{ik}\sin\alpha_{ik} - \frac{\lambda_{ik}M_{gk}}{D}\cos\alpha_{ik}\right)\eta_i\sin f_{bk}$$

According to the relationship between the resultant forces and displacements of the inner ring, the dynamic supporting stiffness matrix of the bearing can be obtained

$$\left[K^b\right] = \begin{bmatrix} \frac{\partial F_x}{\partial \delta_x} & \frac{\partial F_x}{\partial \delta_y} & \frac{\partial F_x}{\partial \delta_z} & \frac{\partial F_x}{\partial \theta_x} & \frac{\partial F_x}{\partial \theta_y} \\ \frac{\partial F_y}{\partial \delta_x} & \frac{\partial F_y}{\partial \delta_y} & \frac{\partial F_y}{\partial \delta_z} & \frac{\partial F_y}{\partial \theta_x} & \frac{\partial F_y}{\partial \theta_y} \\ \frac{\partial F_z}{\partial \delta_x} & \frac{\partial F_z}{\partial \delta_y} & \frac{\partial F_z}{\partial \delta_z} & \frac{\partial F_z}{\partial \theta_x} & \frac{\partial F_z}{\partial \theta_y} \\ \frac{\partial M_x}{\partial \delta_x} & \frac{\partial M_x}{\partial \delta_y} & \frac{\partial M_x}{\partial \delta_z} & \frac{\partial M_x}{\partial \theta_x} & \frac{\partial M_x}{\partial \theta_y} \\ \frac{\partial M_y}{\partial \delta_x} & \frac{\partial M_y}{\partial \delta_y} & \frac{\partial M_y}{\partial \delta_z} & \frac{\partial M_y}{\partial \theta_x} & \frac{\partial M_y}{\partial \theta_y} \end{bmatrix} \tag{17}$$

### 2.1.2. Modeling of Time-Varying Bearing Force

The machining errors of the bearing parts, which has great important effects on the bearing force and dynamic rotational error, are mainly include shape errors, surface waviness and roughness. The surface roughness is not considered in modeling the bearing force for the wavelength of the surface roughness is smaller than the width of the Hertz contact [31].

The outer ring is fixed and the displacements for the inner raceway groove curvature center corresponding to the kth rolling element, including two translations along the *y*-, *z*-axes, and one rotation with respect to the *x*-axis, are expressed as

$$\delta_{ik} = \left[ u_{yk}, u_{zk}, \theta_{xk} \right]^{T} \tag{18}$$

Under the assumption of small deflection, $\delta_{ik}$ could be obtained through coordinate transformations

$$\delta_{ik} = T_k \delta_i + \Delta w_k + \Delta D_k \tag{19}$$

where $T_k$ is the transformation matrix, and can be expressed as

$$T_k = \begin{bmatrix} \sin f_{bk} & \cos f_{bk} & 0 & 0 & 0 \\ 0 & 0 & 1 & \eta_i \cos f_{bk} & -\eta_i \sin f_{bk} \\ 0 & 0 & 0 & \cos f_{bk} & -\sin f_{bk} \end{bmatrix} \tag{20}$$

$\Delta w_k$ denotes the additional displacements caused by the waviness of the bearing rings and rolling elements. Similarly, $\Delta D_k$ represents the additional displacements due to roundness error of the bearing rings and rolling elements.

As illustrated in Figure 4, the waviness exists on each bearing components. $p_{ik}$, $p_{ok}$ and $q_{ik}$, $q_{ok}$ represent radial and axial waviness of inner and outer raceways, $w_{ik}$ and $w_{ok}$ denote the waviness of rolling elements. The harmonic functions are utilized to express the surface waviness due to the waviness vary with time periodically.

$$
\begin{aligned}
p_{ik} &= \sum_{n_{in}=1}^{N} A_{in} \cos\left( n_{in}\left( (\omega_0 - \omega_{bk})t - \tfrac{2\pi(k-1)}{N_b} \right) + f_{in} \right) \\
q_{ik} &= \sum_{n_{in}=1}^{N} B_{in} \cos\left( n_{in}\left( (\omega_0 - \omega_{bk})t - \tfrac{2\pi(k-1)}{N_b} \right) + \psi_{in} \right) \\
p_{ok} &= \sum_{n_{out}=1}^{N} A_{out} \cos\left( n_{out}\left( \omega_{bk}t + \tfrac{2\pi(k-1)}{N_b} \right) + f_{out} \right) \\
q_{ok} &= \sum_{n_{out}=1}^{N} B_{out} \cos\left( n_{out}\left( \omega_{bk}t + \tfrac{2\pi(k-1)}{N_b} \right) + \psi_{out} \right) \\
w_{ik} &= \sum_{n_{out}=1}^{N} C_{ba} \cos(n_{ba}\omega_{sk}t + \varphi_{ba}) \\
w_{ok} &= \sum_{n_{out}=1}^{N} C_{ba} \cos(n_{ba}\omega_{sk}t + \pi + \varphi_{ba})
\end{aligned}
\tag{21}
$$

where $A_{in}$, $A_{out}$, $B_{in}$, $B_{in}$, $C_{ba}$, $C_{ba}$ are the amplitudes of the surface waviness; $n_{in}$, $n_{out}$, $n_{ba}$, are the harmonic orders of the surface waviness; $\phi_{in}$, $\psi_{in}$, $\phi_{out}$, $\psi_{out}$, $\varphi_{ba}$ are the initial phases angle of the surface waviness.

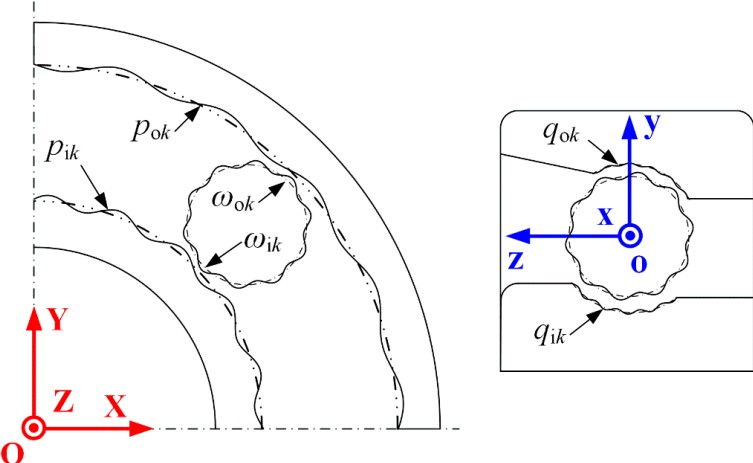

**Figure 4.** The schematic of the bearing waviness.

In accordance with the geometric relations, one can obtain

$$\Delta w_k = \begin{bmatrix} p_{ok} - p_{ik} \\ q_{ok} - q_{ik} \\ 0 \end{bmatrix} \tag{22}$$

The effects of the surface waviness on the dynamic supporting stiffness are very small; however, the effects of the surface waviness on the dynamic rotational error can be reflected in the form of time-varying bearing force. The distances between the inner/outer raceway groove curvature center and the rolling element center, considering the effect of surface waviness, are

$$L_{ik} = (f_i - 0.5)D + p_{ik} - w_{ik}$$
$$L_{ok} = (f_o - 0.5)D + p_{ok} - w_{ok} \tag{23}$$

The geometric relationships can be written as follows:

$$\sin\alpha_{ik} = \frac{L_{ik}\sin\alpha_0 + v_{zk} - u_{zk}}{l_{ik}}$$
$$\cos\alpha_{ik} = \frac{L_{ik}\cos\alpha_0 - v_{yk} + u_{yk}}{l_{ik}}$$
$$\sin\alpha_{ok} = \frac{L_{ok}\sin\alpha_0 - v_{zk}}{l_{ok}}$$
$$\cos\alpha_{ok} = \frac{L_{ok}\cos\alpha_0 + v_{yk}}{l_{ok}} \tag{24}$$

where $v_{yk}$ and $v_{zk}$ are the displacement variation of the $k_{th}$ rolling element along the $y$-axis, $z$- axis, respectively.

The time-varying forces from the kth rolling element to the inner ring considering the surface waviness can be stated as:

$$Q_{ik} = \left[ Q_{yk}, Q_{zk}, Q_{\theta xk} \right] \tag{25}$$

According to Equations (11) and (12), the time-varying force vector $Q_{ik}$ can be expressed as follows:

$$\begin{bmatrix} Q_{yk} \\ Q_{zk} \\ Q_{\theta xk} \end{bmatrix} = \begin{bmatrix} -Q_{ik}\cos\alpha_{ik} + \frac{\lambda_{ik}M_{gk}}{D}\sin\alpha_{ik} \\ Q_{ik}\sin\alpha_{ik} + \frac{\lambda_{ik}M_{gk}}{D}\cos\alpha_{ik} \\ \frac{\lambda_{ik}M_{gk}}{D}r_i \end{bmatrix} \tag{26}$$

Then the sum of the time-varying forces from all of the contacting rolling elements can be obtained:

$$\{F_b\} = \sum_{k=1}^{N_b} \left( [T_k]^T \begin{bmatrix} Q_{yk} \\ Q_{zk} \\ Q_{\theta xk} \end{bmatrix} \right) \tag{27}$$

## 2.2. Dynamic Model of the SRB System

Figure 5 shows the layout of the SRB system. The spindle is supported by two pairs of angular contact bearings. The front and rear bearings are arranged back to back. The types of the front and rear bearing are 7017 and 7016 C, respectively. The light preload is exerted on the SRB system in assembling process. The system coordinate $O\text{-}U_sV_sW_s$ is fixed at the mass center of the spindle. Table 1 shows the parameters of the SRB system. In this section, the dynamic model of the SRB system considering the dynamic supporting stiffness, time-varying bearing force caused by the surface waviness, dynamic unbalance force and the disturbance force, will be derived, which lays a foundation for evaluating the dynamic rotational error.

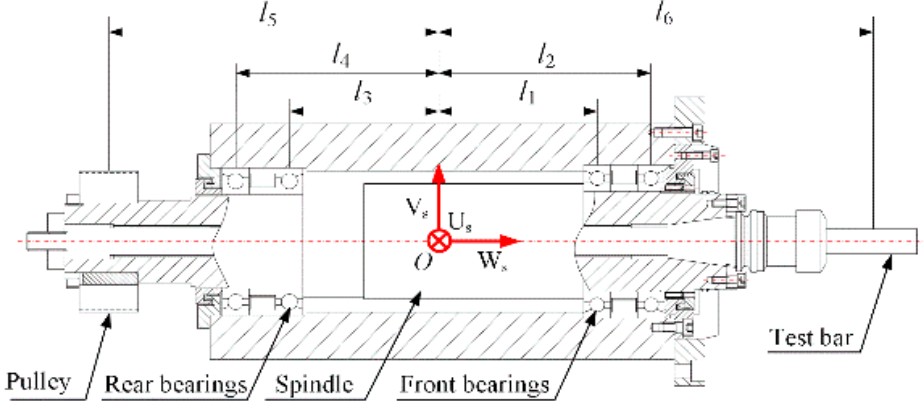

**Figure 5.** Layout of the spindle-rolling bearing (SRB) system.

**Table 1.** Calculation parameters of the SRB system.

| Items | Parameters | Items | Parameters |
|---|---|---|---|
| $m_s$/kg | 30.735 | $l_3$/m | 0.144 |
| $I_s$/kg m$^2$ | 0.963 | $l_4$/m | 0.188 |
| $J_s$/kg m$^2$ | 0.035 | $l_5$/m | 0.312 |
| $l_1$/m | 0.109 | $l_6$/m | 0.446 |
| $l_2$/m | 0.153 | | |

The simplified dynamic model of the SRB system is shown in Figure 6. The spindle can be regarded as a rigid body and has five degrees of freedom, i.e., three translational freedom along the $U_s$-, $V_s$- and $W_s$-axes, and two rotational freedom with respect to the $U_s$- and $V_s$-axes. The dynamic displacement vector of the rotor is

$$q = [u_s, v_s, w_s, \theta_s^u, \theta_s^v]^T \tag{28}$$

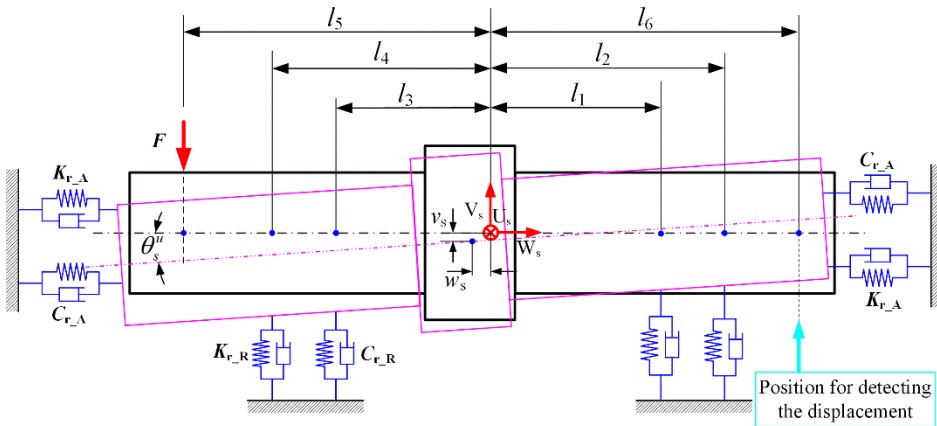

**Figure 6.** Simplified dynamic model of the SRB system.

The dynamic model for SRB system can be obtained based on the Lagrange equation. The final equations of motion of the SRB system can be expressed as

$$[M]\{\ddot{q}\} + ([C] + \omega_0[G])\{\dot{q}\} + [K^b]\{q\} = \{F_e\} + \{F_{dr}\} + \{F_b\} \tag{29}$$

where $[M] = [m_s, m_s, m_s, I_s, I_s]^T$, is the mass matrix of the system, $m_s$ is the mass of the spindle and $I_s$ is the inertia moment of the spindle about the $U_s$-axis and $V_s$-axis. $K^b$ is the stiffness matrix which can be calculated in Section 2. C is the structural damping matrix constructed from modal damping ratios identified experimentally. G is the gyroscopic matrix and can be expressed as

$$[G] = \begin{bmatrix} 0 & 0 & 0 & 0 & 0 \\ 0 & 0 & 0 & 0 & 0 \\ 0 & 0 & 0 & 0 & 0 \\ 0 & 0 & 0 & 0 & -J_s \\ 0 & 0 & 0 & J_s & 0 \end{bmatrix} \tag{30}$$

$J_s$ is the inertia moment of the rotor about the $W_s$-axis. $F_e$ $F_{dr}$ and $F_b$ are the dynamic unbalance force caused by unbalanced mass, disturbance force and the time-varying bearing force caused by surface waviness, respectively.

$F_e$ can be states as

$$\{F_e\} = m_u e_s \omega_0^2 [\cos(\omega_0 t), \sin(\omega_0 t), 0, l_3 \cos(\omega_0 t), l_3 \sin(\omega_0 t)]^T \tag{31}$$

where $m_u$ is the eccentric mass and $e_s$ is the eccentric distance.

The cutting forces have not been considered, since they involve more complicated issues not covered in this paper. The effect of cutting force on the dynamic rotational error will be studied in the future.

### 2.3. Evaluation of Dynamic Rotational Error

It is difficult to measure the dynamic rotational error of the SRB system directly. Precision artifacts are adopted to aid the metrology to identify the dynamic features of the SRB system. Figure 7 shows the diagram of radial error motion measurement and discrete time samples of displacement data measured in radial direction.

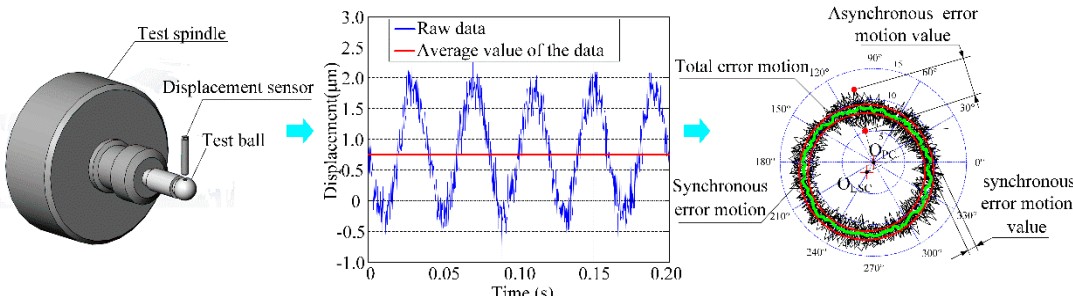

**Figure 7.** Schematic diagram of radial error motion measurement.

The displacement data express sinusoidal trends, with a deviation from zero level. In accordance with the ISO or ANSI/ASME test standard, different kinds of error motion values can be evaluated by using the displacement signals of the artifact [6,7]. The least-square method is used for evaluating the synchronous error motion value. The solution of the synchronous error motion value is transformed into a nonlinear least squares problem. The distances $x_{LSC}$ and $y_{LSC}$ of the least squares center from the polar chart center can be calculated as follows:

$$
\begin{aligned}
x_{LSC} &= \frac{2}{n} \sum_{i=1}^{n} x_i \\
y_{LSC} &= \frac{2}{n} \sum_{i=1}^{n} y_i
\end{aligned}
\tag{32}
$$

where $x_i$ and $y_i$ are the values of the ith data points in the X and Y direction, respectively.

$R_{LSC}$ is the radius the least squares circle and can be calculated as follows:

$$
R_{LSC} = \frac{2}{n} \sum_{i=1}^{n} \sqrt{(x_i - x_{LSC})^2 + (y_i - y_{LSC})^2}
\tag{33}
$$

Error motion polar plots and corresponding error motion values are depicted in the Figure 7.

## 3. Numerical Simulation

Based on the dynamic model established in Section 2 and the ISO 230-7 or ASME B89.3.4 test standard, the influence of different error factors on the dynamic rotational error can be analyzed. Polar plots of the dynamic rotational error can be obtained by using the spindle displacement response, which are measured by the displacement sensor at the end of the SRB system. The least-square method can be used for evaluating the value of synchronous error motion. In this section, the dynamic rotational error of the SRB system has been analyzed quantitatively.

### 3.1. Influence of Bearing Raceway Waviness on Dynamic Rotational Error

Time-varying restoring bearing force can be generated due to the surface waviness of the bearing parts [25]. The excitation caused by bearing waviness is applied on the SRB system. The spindle displacement response is represented as dynamic rotational error when the spindle rotates at high speed. In the simulation, only the effect of a rear bearing waviness is taken into account. Assuming that the bearing inner ring raceway is ideally circular, the rolling elements do not have any shape or dimensional errors. The outer raceway surface has a single order of radial and axial sinusoidal waviness. The amplitude of the bearing waviness is 1–2 μm, and the harmonic order is 11–20 UPR (Undulations Per Revolution). To show the relationships more distinctly between radial errors and the number of undulations and rolling elements, values that are larger than actual errors are selected. The number of rear bearing rolling elements is 20. The length of numerical simulation is set to

20 revolutions. The discrete time response signals and the polar plots of the radial error motion are shown in Figure 8.

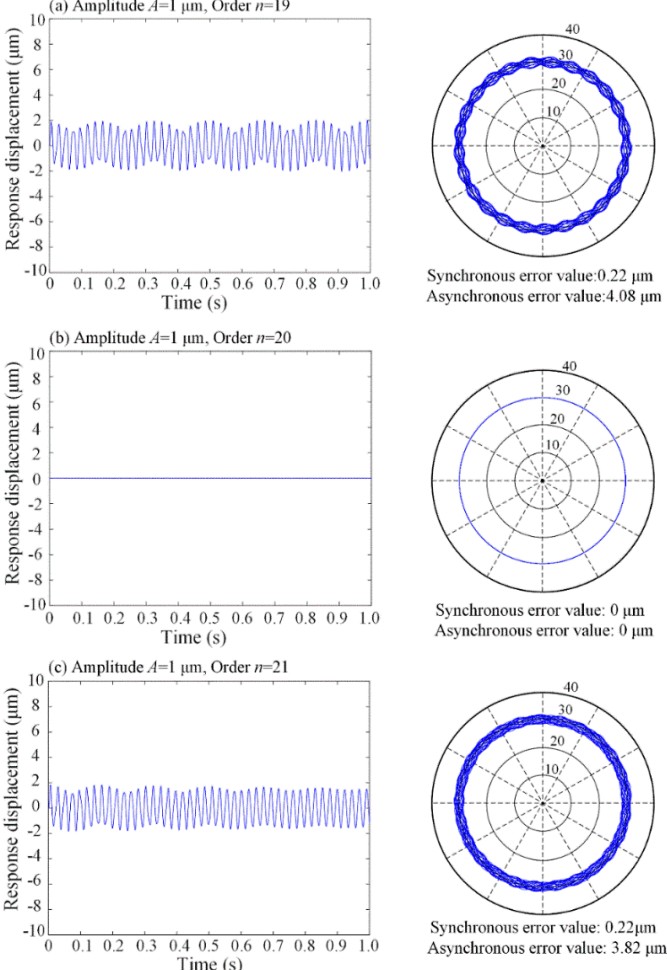

**Figure 8.** Responses and rotational errors with the amplitude (1 μm) and different orders of the surface waviness.

As can be seen from Figures 8 and 9, when the bearing waviness has the same amplitude, the radial displacement response in the $V_s$-axis is related to the order of the bearing waviness. When the order of the waviness and the number of rolling elements are the same, the waviness has almost no effect on the rotational error motion. However, when the difference between the order and the number of rolling elements is 1, a very obvious asynchronous error motion can be generated in the SRB system. Meanwhile, waviness hardly causes synchronous error motion. The same trends can be obtained by comparing the influence of the waviness on the rotational error of a single bearing in the literatures [32,33]. In the comparing literature, an angular contact ball bearing (SKF, 7012/CD) is chosen for the calculation. The comparison of the effects on radial error motion is shown in Figure 10.

When the order of surface waviness is the same, the magnitude of the bearing waviness determines the magnitude of the rotational error. The rotational error motion value, especially for asynchronous error, increases with the increase of the waviness amplitude.

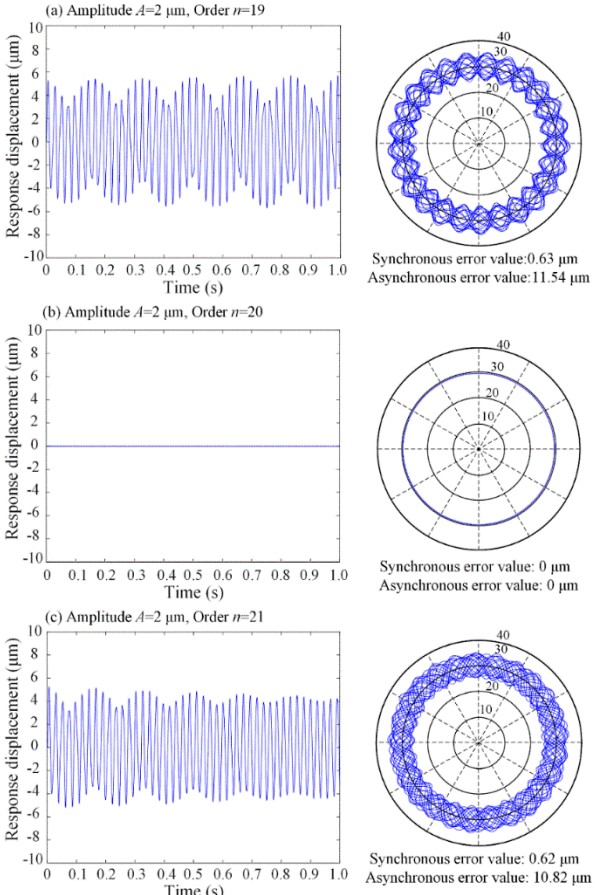

**Figure 9.** Responses and rotational errors with the amplitude (2 μm) and different orders of the surface waviness.

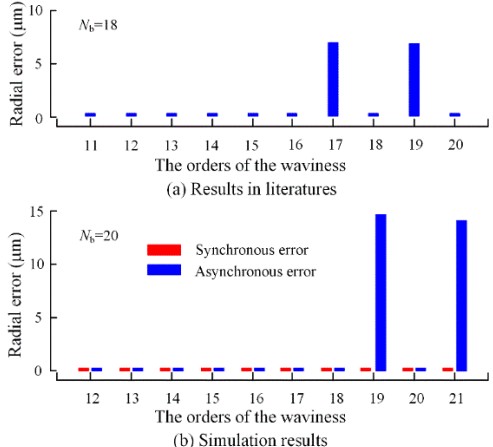

**Figure 10.** Compared with the results in the literatures.

### 3.2. Influence of Dynamic Unbalance on Dynamic Rotational Error

As a typical rotary machine, dynamic unbalance force is a critical factor attributed to the error motions of the SRB system when its center of mass deviates from the axis of rotation. As a kind of dynamic excitation, the frequency of the dynamic unbalance force is the same as the rotation frequency of the spindle. In the traditional test of the spindle rotational error motion, the fundamental frequency components, including installation eccentricity error of the target, are filtered. The dynamic unbalance force is a function of the rotational speed. From dynamic point of view, the vibration caused by

dynamic unbalance force directly affects the total indicator reading (TIR) of the SRB system. Assume that the spindle has an eccentric mass $m_u$ = 0.005 kg on the pulley side, and the eccentricity $e_c$ = 0.054 m. The spindle rotational speed ranges from 1000 to 10,000 rpm. The response displacements at the front end of the SRB system are shown in Figure 11.

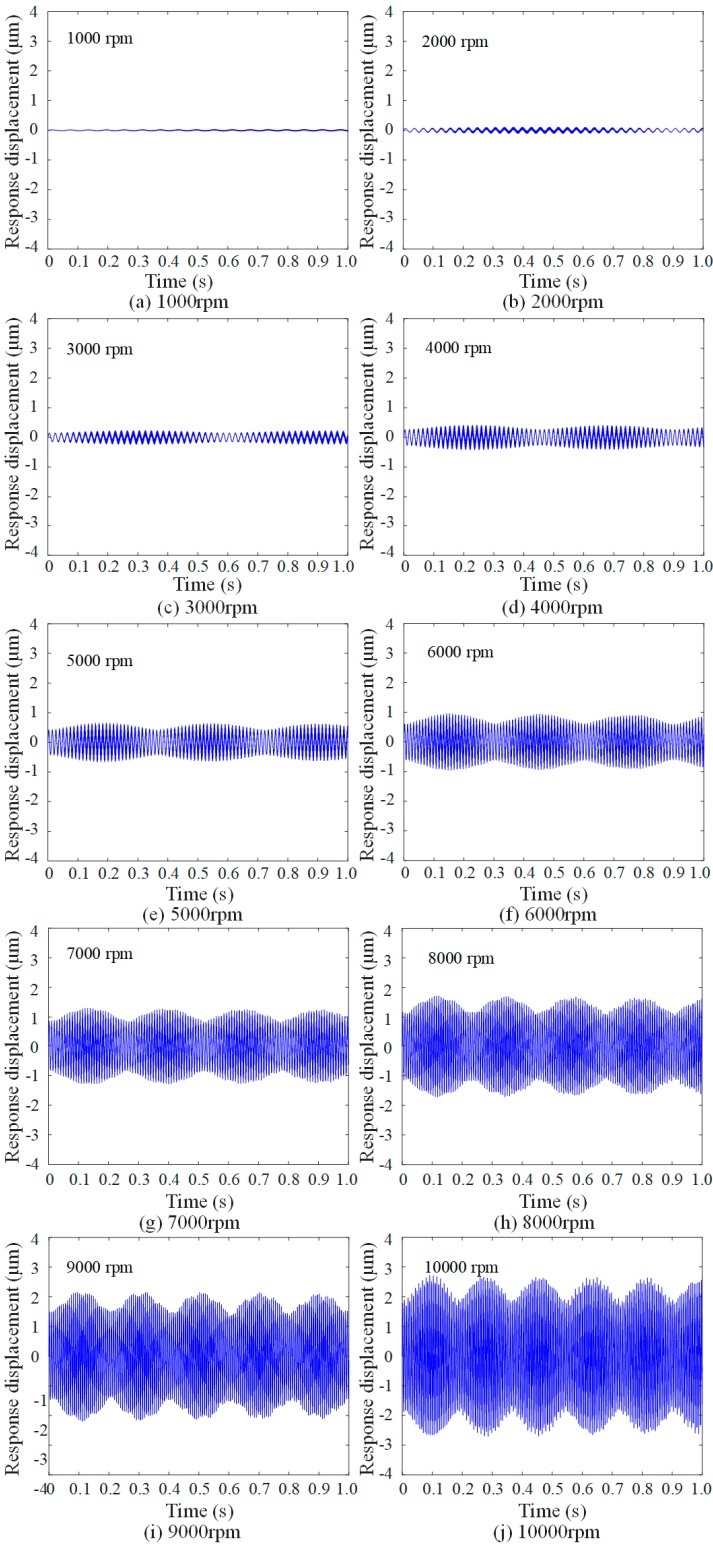

**Figure 11.** Effects of dynamic unbalanced force on total indicator reading (TIR) at different speeds.

The trends of the overall effect of spindle unbalance on TIR, synchronous and asynchronous errors are shown in Figure 12. As can be seen in Figure 12, the TIR caused by the spindle unbalance increases from 0.07 μm in 1000 rpm to 5.61 μm in 10,000 rpm. Spindle unbalance has little effect on the synchronous error, and the overall synchronous error is less than 0.4 μm. The asynchronous error increases from 0.03 to 2.27 μm, as the speed increasing from 1000 to 10,000 rpm.

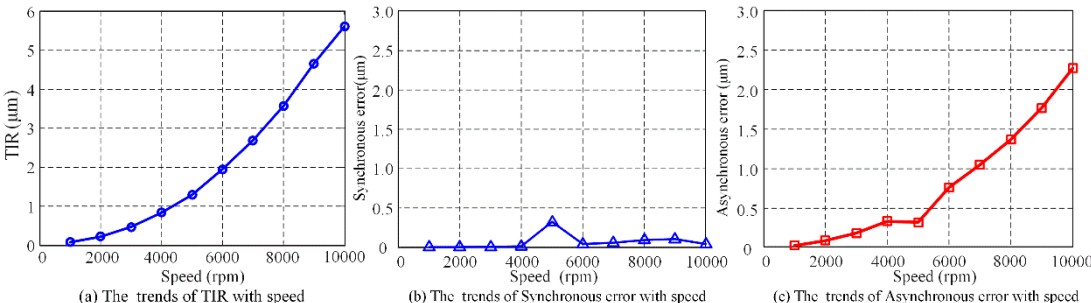

**Figure 12.** Trends of impact of dynamic unbalance force on TIR, synchronous/asynchronous errors.

The frequency domain analysis of the simulated displacement response at 10,000 rpm is shown in Figure 13.

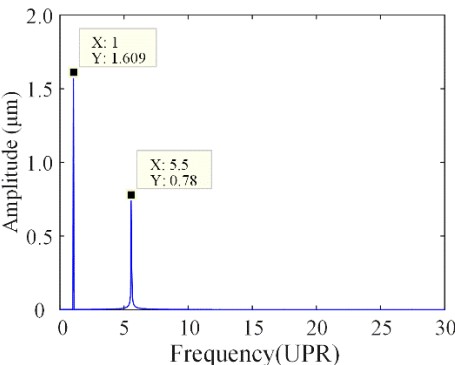

**Figure 13.** Frequency analysis of response displacement at 10,000 rpm.

The frequency domain analysis shows that it mainly includes two peaks. The frequency of the first peak corresponds to the excitation frequency of the spindle dynamic unbalance force, that is, the fundamental frequency (spindle rotational frequency). The frequency of the second peak corresponds to the spindle resonance frequency (natural frequency of the SRB system). The asynchronous error motion presents large value due to the resonance frequency is non-integer multiples of the fundamental frequency.

### 3.3. Influence of Disturbance Force on Dynamic Rotational Error

The spindle is rotated by the drive system. The mechanical spindle often uses a wedge belt or a timing belt drive, while the electric spindle is driven by built-in motor. During the rotation of the spindle, the transmission system and the electromagnetic force will generate a disturbance force on the SRB system. The dynamic response of the SRB system to disturbance forces is also reflected in the dynamic rotational error. Based on the established model, the influence of disturbance force on dynamic rotational error can be analyzed numerically. Assuming that the frequency of the disturbance force is an integer multiple of the fundamental frequency, the frequency order ranges from 2 to 10 UPR. Polar plots of the dynamic rotational error, which are caused by the disturbance force with the amplitude of 200 N, are shown in Figure 14.

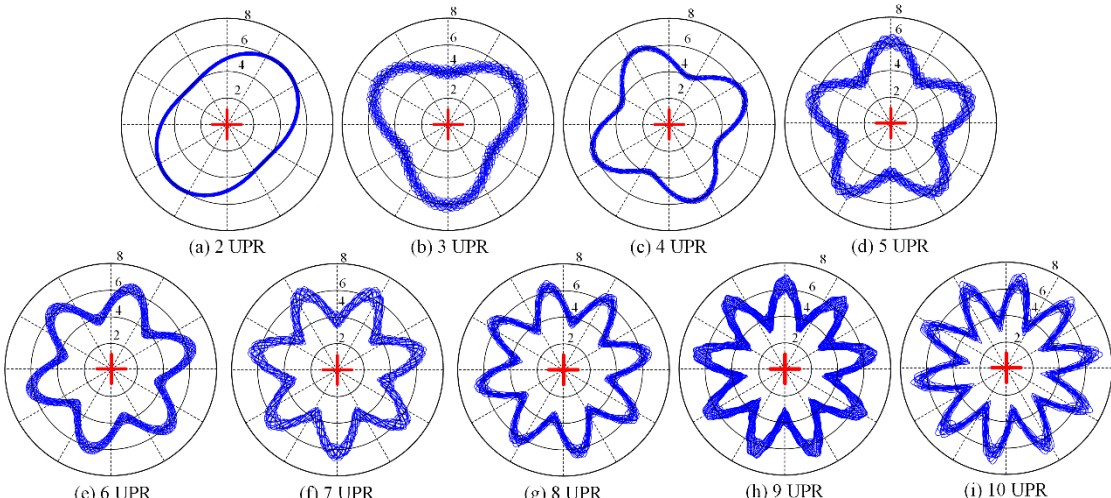

**Figure 14.** Effects of external disturbance force on radial synchronous errors.

As shown in the polar plots, the influence of external disturbance force, such as the drive system, on the dynamic rotational error is mainly forced vibration, and the frequency of the radial synchronous error is the same as the frequency of the disturbance force. By increasing the amplitude of the disturbance force, the comparison results of the dynamic rotational errors of different orders and amplitudes can be obtained, as shown in Figure 15.

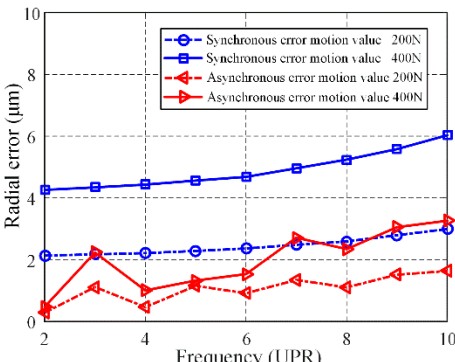

**Figure 15.** Variation trends of radial errors with different amplitude and order of external disturbance force.

The simulation result shows that both synchronous and asynchronous error motion values increase slowly with the increasing of disturbance force order. When increasing the amplitude of the disturbance force from 200 to 400 N, the range of synchronous error motion values increase from 2.13–2.99 to 4.26–6.03 μm, and the range of asynchronous error motion values increase from 0.28–1.63 to 0.47–3.26 μm. Both the synchronous error and the asynchronous error increase as the amplitude of the disturbance force increases.

## 4. Experimental Validation

The experimental platform is built up to further verify the validity of the established model. Figure 16 shows the photo of the experimental setup. A mechanical machine tool spindle, which is driven by belt driving system, is applied to explore these issues. A high-precision dual ball artifact mounted in the rotor is used as the testing target, through which eccentricity can be adjusted manually. The NI (National Instruments) data acquisition (DAQ) card (USB 6356) and capacitive displacement sensor with nanometer-level resolution are used to acquire the displacement signals. Table 2 shows the technical parameters of the capacitive sensors used in the experiment. Data acquisition and processing

software is developed by LabVIEW software and can be used for online analysis. The displacement data can be also saved for off-line processing.

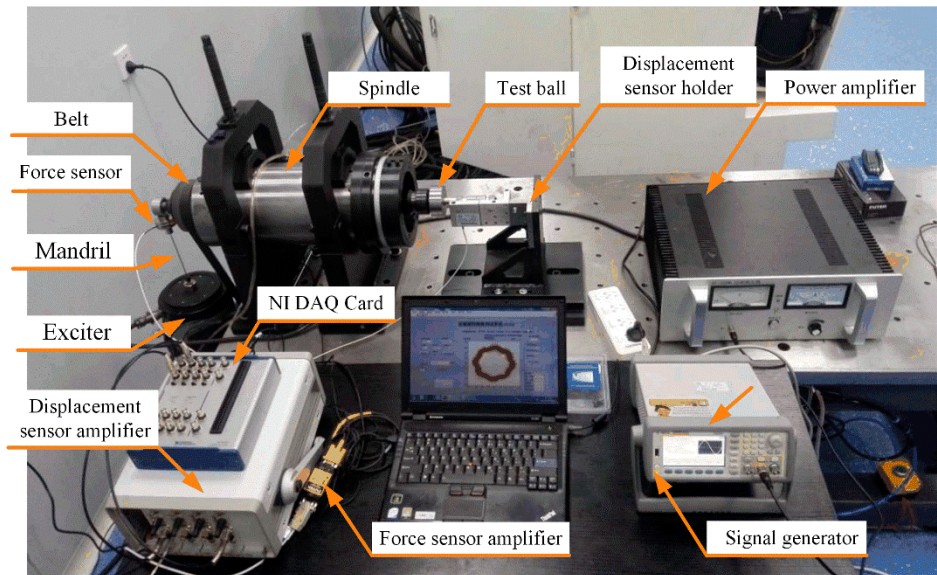

**Figure 16.** Test setup of disturbance force verification experiment.

**Table 2.** Technical parameters of capacitive sensor.

| Item | Parameters | Item | Parameter |
|---|---|---|---|
| Sensing area diameter | 2/mm | Linearity error | 0.1% |
| Measurement range | 250/μm | Operating temperature | 4–50/°C |
| Output voltage | ±10/V | Output sensitivity | 0.08/V/μm |
| Connection mode | Differential | Ambient temperature | 20 ± 1/°C |

In order to investigate the dynamic rotational error quantitatively, two experiments are designed. The schematic diagram of unbalanced mass and disturbance force verification experiment is illustrated in Figure 17. In the actual operation of the SRB system, there are many factors that affect its dynamic rotational error. It is difficult to eliminate the impact of these factors effectively. To study the influence of a certain factor, the proportion of this factor can be artificially enlarged. Compared with other factors, it will produce more obvious changes.

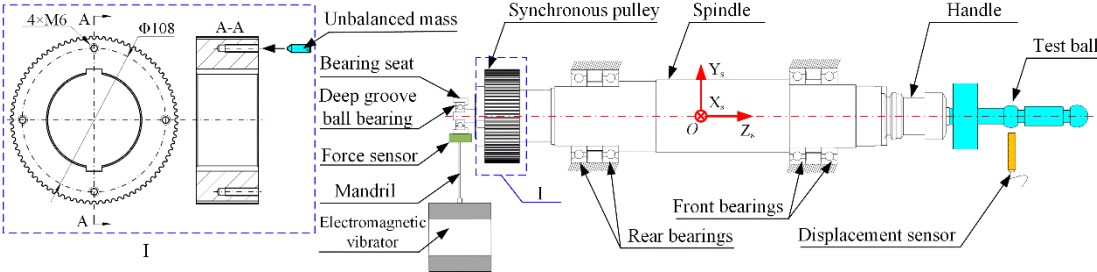

**Figure 17.** Schematic diagram of unbalanced mass and disturbance force verification experiment.

When the SRB system is not subjected to any artificial excitation force, rotational error motion can also occur due to the influence of some unknown factors. The spindle rotational error measured in this state is used as the initial value for comparison. The polar plot is shown in Figure 18. The whirling speed is 300 rpm. The corresponding TIR, synchronous and asynchronous errors are 14.41, 0.55 and 2.86 μm, respectively.

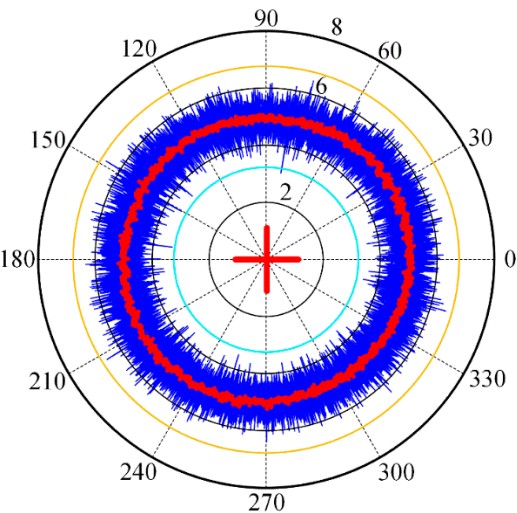

**Figure 18.** Spindle rotational error of the initial state.

### 4.1. Validation of Dynamic Unbalance Force

The spindle rotor used in the experiment is handled by dynamic balancing and it produces very little centrifugal force at high speeds. An unbalanced mass is mounted at the location of pulley to generate dynamic unbalance force when it rotates, as shown in Figure 17. The dynamic unbalance force generated by unbalanced mass increases as the increasing of whirling speed. The dynamic rotational error can be measured by the DAQ system.

In the experiment, the whirling speed of SRB system ranges from 300 to 6000 rpm. The length of sampling is set at 60 revolutions, with the sampling frequency of 15 kHz. The dynamic unbalance force generated at 300 rpm is about 0.54 N. The TIR, synchronous and asynchronous errors are selected as the initial value due to some unknown factors except unbalance force. The rotational errors generated in the process of increasing the speed are all considered to be caused by the dynamic unbalance force. The measured TIR, synchronous and asynchronous errors under different speeds, as well as the comparisons with the simulated values, are shown in Figure 19.

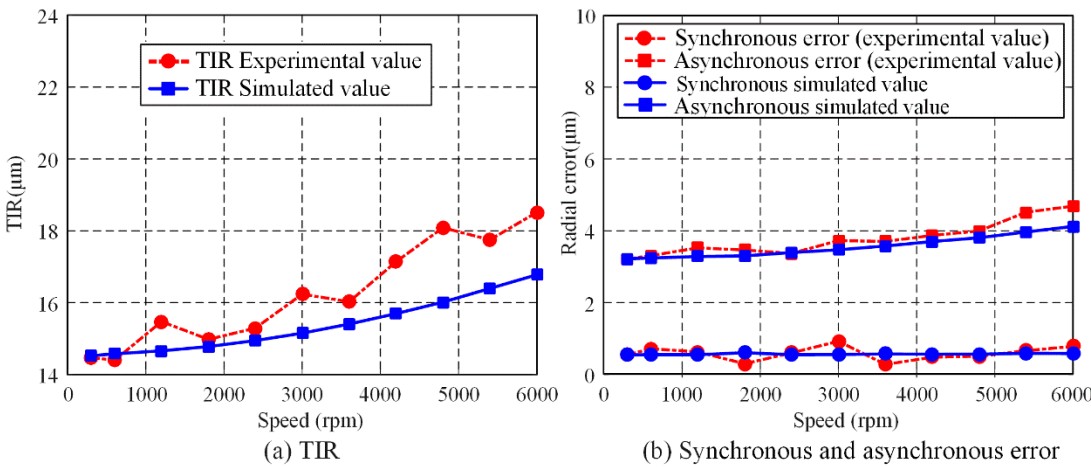

**Figure 19.** Comparing the experimental results with the simulated values.

It can be seen from Figure 19a that experimental results agree well with the simulated values. The simulated value of TIR varies from 14.53 μm at 300 rpm to 16.78 μm at 6000 rpm, while the experimental value varies from 14.41 μm to 17.91 μm. The difference between the simulated and experimental TIR values become large from 3000 to 6000 rpm, due to the vibration of transmission system. Figure 19b indicates that dynamic unbalance force basically has no influence on synchronous

error, which is consistent with the simulated results. However, dynamic unbalance force has a significant effect on asynchronous errors. Asynchronous errors due to unbalanced mass increase with the speed increasing. The experimental value of asynchronous error varies from 3.21 μm at 300 rpm to 4.69 μm at 6000 rpm, while the simulated value varies from 3.22 to 4.12 μm.

### 4.2. Validation of Disturbance Force

The same idea is adopted in this section that the disturbance force is enlarged artificially. External radial dynamic loading is applied by using a signal generator, power amplifier and electromagnetic vibrator [34]. A signal with known frequency is generated by the signal generator and amplified by a power amplifier to drive the electromagnetic vibrator to generate a dynamic disturbance force. A deep groove ball bearing, which is utilized to apply dynamic radial force to the SRB system, is mounted on the end of the rotor [34]. The error inside the load bearing can be omitted. A force sensor is used to measure the disturbance force in time domain. The schematic diagram and the experimental setup are shown in Figure 17.

In order to eliminate the influence of whirling speed change, the speed is set at 300 rpm. The amplitude and frequency of disturbance force are changed manually. The sinusoidal signal generated by the signal generator ranges from 10 to 40 Hz. The amplitude of the disturbance force can be adjusted by using the voltage gain of the power amplifier, approximately between 50 to 100 N. The theoretical values of the dynamic rotational error caused by the disturbance force are calculated by using the actual values measured by the force sensor. The spindle synchronous errors of different exciting orders are shown in Figure 20.

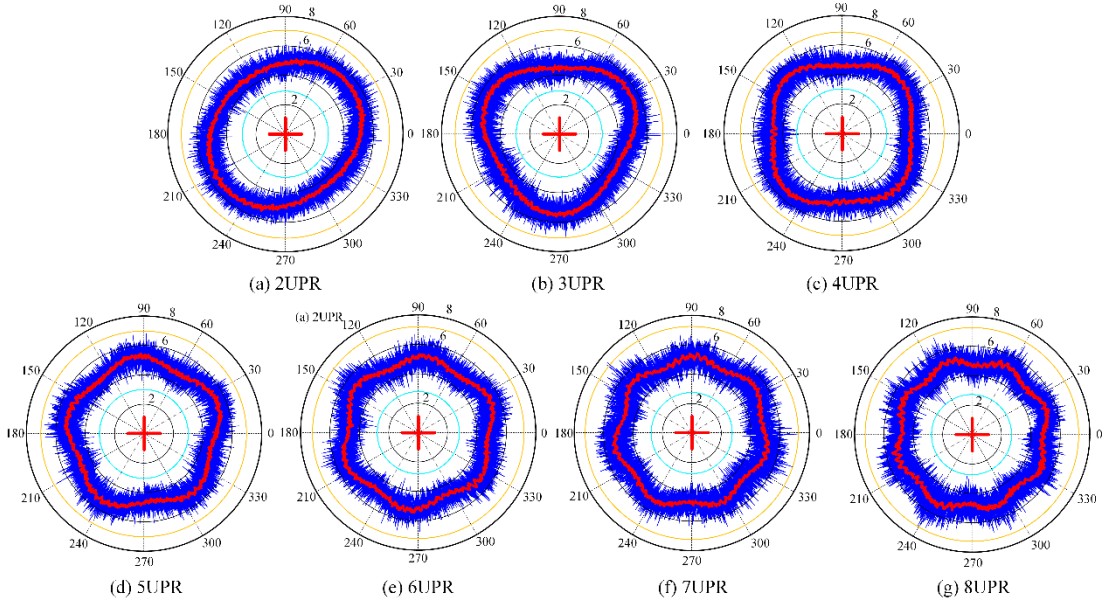

**Figure 20.** Spindle synchronous errors induced by different order of disturbance force.

By changing the amplitude of the disturbance force, the synchronous and asynchronous errors caused by the disturbance force with different amplitudes and frequencies can be obtained, as shown in Figure 21. It can be seen that the amplitude of the disturbance force decreases as the increasing of frequency. So, the corresponding synchronous error also shows an overall downward trend. However, the asynchronous error does not take obvious change. This can be attributed to the fact that the range of disturbance force is not large enough. The experimental values of different amplitude forces are 1.39–1.02 and 1.82–1.32 μm, respectively. The corresponding theoretical values are 1.22–1.03 and 1.52–1.29 μm, respectively. The maximum difference occurs at 3 UPR for about 0.3 μm. The comparison indicates that the experimental values agree with the theoretical values.

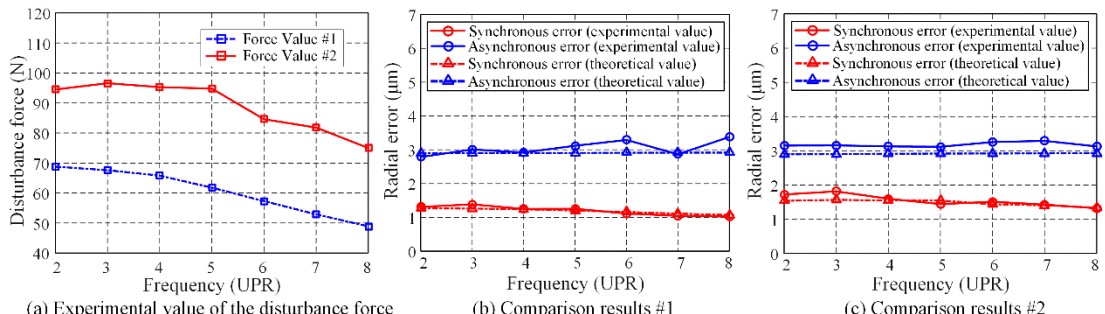

**Figure 21.** Spindle synchronous errors induced by different amplitude of disturbance force.

## 5. Conclusions

An analytical modeling method, based on dynamic of the SRB system which considers the bearing nonlinearity, is presented in this paper, so as to study the formative mechanism of dynamic rotational error and predict the rotational performance for the SRB system. Core conclusions of the study as a whole are as follows.

The presented modeling method for dynamic rational error, based on the dynamic model of the SRB system, is feasible and valid to analyze the formative mechanism of the rotational error motion. Surface waviness, dynamic unbalance force and disturbance force are involved in the established model. The surface waviness hardly causes synchronous error motion. However, when the difference between the order and the number of rolling elements is one, the SRB system would produce a very obvious asynchronous error motion. The effects of dynamic unbalance and disturbance force are verified by the comparison experiments. Dynamic unbalance force has little effect on the synchronous error; however, the TIR and asynchronous error increases with the increasing of rotational speed and unbalance mass. Both synchronous and asynchronous error motion values increase when the amplitude of disturbance force and its frequency increases. Besides, the frequency of the radial synchronous error is the same as that of the disturbance force. The merit of this modeling method is that it can provide designers and/or field engineers with an informative guideline to improve the dynamic rotational accuracy.

**Study prospects:**

Based on the modeling method in this paper, investigations about the effect of temperature and lubrication on the dynamic rotational error during the spindle operation will be emphasized in future studies.

**Author Contributions:** Conceptualization, Y.T., H.Q., and G.H.; Funding acquisition, G.J., L.H. and T.W.; Investigation, G.H. and Y.C.; Methodology, G.H. and Y.C.; Project administration, Y.T., H.Q.; Supervision, L.C. and L.H.; Validation, L.C. and L.H.; Writing—original draft, G.H.; Writing—review and editing, G.H. and Y.C. All authors have read and agreed to the published version of the manuscript.

**Funding:** The authors gratefully acknowledge the support of the National Science Foundation of China No.51705364, the fund of Tianjin Natural Science Foundation (18JCZDJC10050, 20JCQNJC01060 and 18JCQNJC74900), the fund of Tianjin Municipal Education Commission (Grant No. JWK1601).

**Conflicts of Interest:** The authors declare no conflict of interest. The funders had no role in the design of the study; in the collection, analyses, or interpretation of data; in the writing of the manuscript, or in the decision to publish the results.

## Nomenclature

| | |
|---|---|
| $\delta_i$ | The displacement vector of the inner ring |
| $\phi_{bk}$ | The position angle of the kth ball |
| $N_b$ | Number of the rolling element of the bearing |
| $\omega_0$ | The rotation angular speed of the inner ring (rad/s) |
| $\omega_{bk}$ | The orbital angular speed of the rolling element (rad/s) |
| $\omega_{sk}$ | The self-rotation angular speed of the rolling element (rad/s) |
| $\alpha_o$ | Initial contact angle of the bearing (°) |

| | |
|---|---|
| $\alpha_{ik}$ | Inner ring contact angle of the bearing of the kth rolling element (°) |
| $\alpha_{ik}$ | Outer ring contact angle of the bearing of the kth rolling element (°) |
| $\beta_k$ | The angle between the spinning axis of the rolling elements and Z axis (°) |
| $D$ | The diameter of the rolling element (mm) |
| $d_m$ | Pitch Diameter (mm) |
| $\delta_{ik}/\delta_{ok}$ | The contact deformation between the inner/outer bearing ring and kth rolling element |
| $f_i/f_o$ | Inner/Outer groove curvature radius coefficient |
| $r_i/r_o$ | Inner/Outer groove curvature radius (mm) |
| $\eta_i$ | The distance between the rotating axis Z and inner raceway groove curvature center (mm) |
| $L_{ik}/L_{ok}$ | The distance between the inner/outer raceway groove curvature center and the rolling element center before deformation |
| $l_{ik}/l_{ok}$ | The distance between the inner/outer raceway groove curvature center and the rolling element center after deformation |
| $Q_{ik}/Q_{ok}$ | The contact force between the inner/outer ring and the rolling element (N) |
| $M_{gk}$ | Gyroscopic moment of the kth rolling element (N.mm) |
| $K_{ok}$ | Coefficient of the load-displacement between inner ring and kth rolling element (N/mm1.5) |
| $K_{ik}$ | Coefficient of the load-displacement between outer ring and kth rolling element (N/mm1.5) |
| $\lambda_{ik}, \lambda_{ok},$ | The friction coefficients between the inner/outer raceway and kth rolling element |
| $F_{ck}$ | The centrifugal force of kth rolling element (N) |
| $F_i$ | The resultant force vector of the inner ring due to all of the contacting rolling elements (N) |
| $K_b$ | The dynamic supporting stiffness matrix of the bearing (N/μm) |
| $\delta_{ik}$ | The displacements for inner raceway groove curvature center corresponding to the kth rolling element (mm) |
| $T_k$ | The transformation matrix |
| $\Delta \mathbf{w}_k$ | The additional displacements caused by the waviness of the bearing rings and rolling elements (mm) |
| $\Delta \mathbf{D}_k$ | The additional displacements due to roundness error of the bearing rings and rolling elements (mm) |
| $p_{ik}, p_{ok}$ | The radial waviness of inner and outer raceways |
| $p_{ik}, p_{ok}$ | The axial waviness of inner and outer raceways |
| $w_{ik}, w_{ok}$ | The waviness of kth rolling elements |
| $A_{in}, A_{out}, B_{in},$ $B_{in}, C_{ba}, C_{ba}$ | The amplitudes of the surface waviness |
| $n_{in}, n_{out}, n_{ba},$ | The harmonic orders of the surface waviness |
| $\phi_{in}, \psi_{in}, \phi_{out},$ $\psi_{out}, \varphi_{ba}$ | The initial phases angle of the surface waviness |
| $F_b$ | The sum of the time-varying forces from all of the contacting rolling elements (N) |
| $q$ | The dynamic displacement vector of the rotor |
| $\alpha_i$ | Outer ring contact angle of the bearing (°) |
| $C$ | The structural damping matrix of the system |
| $G$ | The gyroscopic matrix of the system |
| $J_s$ | The inertia moment of the rotor about the $W_s$-axis |
| $R_{LSC}$ | The radius the least squares circle |

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
