# Peer review of "Investigation on Modeling and Formation Mechanism of Dynamic Rotational Error for Spindle-Rolling Bearing System"

_applsci, doi:10.3390/app10175753_

Round 1

Reviewer 1 Report

The authors explored the modeling method and formative mechanism of the dynamic rotational error for the SRB system using both experimental and numerical modeling. I'm not an expert in SRB systems. But I have a few suggestions that can help to improve the paper. 

-what's this part in the introduction?

"The introduction should briefly place the study in a broad context and highlight why it is 36 important. It should define the purpose of the work and its significance. The current state of the 37 research field should be reviewed carefully and key publications cited. Please highlight controversial ....."

-Authors should cite other works anywhere any famous physics-based or mechanics-based equation/formulation is used that is not author's work.

-In Figure 10. (Compared with the results in the litreture), it's not clear how the the comparison was done. More details needed to show how the comparison done. what's N_b?

"Dynamic unbalance force has little effect on the synchronous  error, however, the TIR and asynchronous error increases with the increasing of rotational speed and unbalance mass" is this consistent with the results of other research efforts in the litreture?

Author Response

Dear Reviewer:

Thank you for your comments concerning our manuscript entitled “Investigation on modeling and formation mechanism of dynamic rotational error for spindle-rolling bearing system” (ID: APPLSCI-889327). Those comments are all valuable and very helpful for revising and improving our paper, as well as the important guiding significance to our researches. We have studied comments carefully and have made corrections which we hope meet with approval. All changes are highlighted in the revised manuscript. Below are the point-by-point responses to the comments of the reviewers:

Reviewer #1

The authors explored the modeling method and formative mechanism of the dynamic rotational error for the SRB system using both experimental and numerical modeling. I'm not an expert in SRB systems. But I have a few suggestions that can help to improve the paper.

1-what's this part in the introduction?

"The introduction should briefly place the study in a broad context and highlight why it is important. It should define the purpose of the work and its significance. The current state of the research field should be reviewed carefully and key publications cited. Please highlight controversial ....."

[Authors’ Response]:

Thanks for your comments. We are so sorry for the formatting error. The instructional text will be deleted. Besides, we rewrite the introduction and try to make it clear and concise.

2-Authors should cite other works anywhere any famous physics-based or mechanics-based equation/formulation is used that is not author's work.

[Authors’ Response]:

In order to illustrate the modeling method and formative mechanism of the rotational error motions for the SRB system rigorously and repeatability,some equations/formulations are referenced by any other literature. All the cited equation/formulation will be marked in the revised manuscript. Thanks for your suggestion.

3-In Figure 10. (Compared with the results in the litreture), it's not clear how the the comparison was done. More details needed to show how the comparison done. what's N_b?

[Authors’ Response]:

Thanks for your comments.  As can be seen from the Figure 10,when the difference between the order and the number of rolling elements is 1, a very obvious asynchronous error motion can be generated in the SRB system. Meanwhile, waviness hardly causes synchronous error motion. This conclusion hardly can be verified by experiment,However, the same conclusion can be obtained by the literatures [32,33] and the method we used is different form theirs. To some extent, it verifies the correctness of our conclusion. The meaning of comparison with the results in the literature is that we hope our paper will be of particular help to the study of dynamic rotational error.

The symbol“N_b”is the number of the rolling element of the bearing, The symbol“n”is  the order of the bearing waviness.

[32]  Liu, J.; Hong, J.; Zhu, Y. S.; Li, X. H.; Running accuracy of the high speed precision angular contact ball bearings. Acad. J. XiAn Jiaotong Univ. 2011, 45, 72-78.

[33]  Noguchi, S.; Tanaka, K. Theoretical analysis of a ball bearing used in HDD spindle motors for reduction of NRRO. IEEE T. Magn. 1999, 35, 845-850.

4-"Dynamic unbalance force has little effect on the synchronous error, however, the TIR and asynchronous error increases with the increasing of rotational speed and unbalance mass" is this consistent with the results of other research efforts in the litreture?

[Authors’ Response]:

Thanks for your comments. This conclusion, which is not mentioned in any other literature, is one of the innovations of this manuscript. The effects of dynamic unbalance force on the synchronous error and asynchronous error of the spindle is explored both theoretically and experimentally. Experimental setup is established based on a typical SRB system and a series of experiments are carried out. The experimental results are in good agreement with the theoretical and simulation results, which can demonstrate the feasibility and validity of the obtained conclusions.

We appreciate for reviewers' warm work earnestly, and hope that the correction will meet with approval. Once again, thank you very much for your comments and suggestions.

Reviewer 2 Report

The manuscript discusses a dynamic model of the SRB system considering the bearing nonlinearity. Given below are my comments to improvise upon the quality of the manuscript.

  1. The first paragraph of the introduction makes no sense, it seems that the authors have forgotten to delete the instructions from the manuscript template. Kindly remove lines 36-43.
  2. In line 111, add a brief explanation for the hertzian contact theory.
  3. Please correct the typos in ωo. The symbol ‘o’ should be in the subscript. Also, please check for other such existing typos throughout the manuscript.
  4. It is rather unclear to me that which of the equations are developed by the authors, and which of them are taken from the literature. All the equations from the literature should be properly referenced.
  5. In line 246, please explain why it is difficult to measure the dynamic rotational error of the SRB system.

Author Response

Dear Reviewer:

Thank you for your comments concerning our manuscript entitled “Investigation on modeling and formation mechanism of dynamic rotational error for spindle-rolling bearing system” (ID: APPLSCI-889327). Those comments are all valuable and very helpful for revising and improving our paper, as well as the important guiding significance to our researches. We have studied comments carefully and have made corrections which we hope meet with approval. All changes are highlighted in the revised manuscript. Below are the point-by-point responses to the comments of the reviewers:

Reviewer #2

The manuscript discusses a dynamic model of the SRB system considering the bearing nonlinearity. Given below are my comments to improvise upon the quality of the manuscript.

1-The first paragraph of the introduction makes no sense, it seems that the authors have forgotten to delete the instructions from the manuscript template. Kindly remove lines 36-43.

[Authors’ Response]:

Thanks for your precious opinions. We are so sorry for the formatting error. The instructional text will be deleted. Besides, we rewrite the introduction and try to make it clear and concise.

2-In line 111, add a brief explanation for the hertzian contact theory.

[Authors’ Response]:

Thanks for your precious suggestion. In order to make the concepts in this article more relevant and easy to comprehend, a brief explanation for the Hertz contact theory will be added in the revised manuscript.  

3-Please correct the typos in ωo. The symbol ‘o’ should be in the subscript. Also, please check for other such existing typos throughout the manuscript.

[Authors’ Response]:

Thanks for your comments. The manuscript has been revised according to reviewer’s comments. We have checked the paper thoroughly and corrected some mistakes in it.

4-It is rather unclear to me that which of the equations are developed by the authors, and which of them are taken from the literature. All the equations from the literature should be properly referenced.

[Authors’ Response]:

Thanks for your comments. In order to illustrate the modeling method and formative mechanism of the rotational error motions for the SRB system rigorously and repeatability,some equations/formulations are referenced by any other literature. All the cited equation/formulation will be marked in the revised manuscript. Thanks for your suggestion.

5-In line 246, please explain why it is difficult to measure the dynamic rotational error of the SRB system.

[Authors’ Response]:

Thanks for your comments. It is difficult to measure the dynamic rotational error of the SRB system directly. The dynamic rotational error is the error motion of the axis of the rotor (Ref. [6,7]) . The dynamic rotational error of the SRB system is evaluated through a certain algorithm according to the measured data.

[6] ANSI/ASME. B89.3.4. Axes of rotation: methods for specifying and testing. Publisher: ASME,New York, USA, 2010; pp. 35–36.

[7] ISO 230-7. Test code for machine tools. Part 7: geometric accuracy of axes of rotation. Publisher: International Standardization Organization, Geneva, Switzerland, 2006; pp. 16–19.

We appreciate for reviewers' warm work earnestly, and hope that the correction will meet with approval. Once again, thank you very much for your comments and suggestions.

Round 2

Reviewer 2 Report

Thank you for taking my comments into account and updating the manuscript accordingly. The quality of the manuscript has now improved considerably.